# Control of Dielectric, Mechanical, and Thermal Properties of a Polymer Composite Based on ABS Using CoB Nanoparticles

**DOI:** 10.3390/polym17010038

**Published:** 2024-12-27

**Authors:** Artur Khannanov, Anastasia Burmatova, Dinar Balkaev, Anastasia Rossova, Konstantin Zimin, Airat Kiiamov, Mikhail Cherosov, Ivan Lounev, Marianna Kutyreva

**Affiliations:** 1A.M. Butlerov Chemical Institute, Kazan Federal University, Kazan 420008, Russia; anaeburmatova@kpfu.ru (A.B.); dinar.balkaev@yandex.ru (D.B.); anastasia.rossova@yandex.ru (A.R.); kostzim@list.ru (K.Z.); mkutyreva@mail.ru (M.K.); 2Institute of Physics, Kazan Federal University, Kazan 420008, Russia; ajrat.kiyamov@kpfu.ru (A.K.); miacherosov@kpfu.ru (M.C.); lounev75@mail.ru (I.L.)

**Keywords:** cobalt boride, composite materials, ABS plastic, glass transition temperature, superparamagnetic composites, dielectric constant

## Abstract

This article is devoted to the development of a new method for the synthesis of magnetic cobalt boride nanoparticles using a low-energy approach. The obtained nanoparticles were used to create composite materials based on industrial thermoplastic ABS. The effect of different concentrations of nanoparticles on the physical, mechanical, magnetic, and dielectric properties of composite materials was studied. It was proven that by varying the concentration of nanoparticles in the ABS composite, it is possible to control the glass transition temperature from 107.5 to 112 °C. The resulting composites demonstrated superparamagnetic behavior, which changed linearly. The permittivity of the composite remained close to that of pure ABS, but a shift in the maximum permittivity to the low-frequency region was observed with an increase in the content of nanoparticles. Thus, a method for controlling the mechanical, magnetic, and dielectric properties of a composite material has been developed, which makes it possible to use routine ABS in a wider range of applications, including electrical devices.

## 1. Introduction

The creation of functional polymers and composite materials is one of the key areas of modern materials science. Increasingly, attention is paid to highly specialized polymeric materials with a certain functionality, [1,2,3] which, in the future, will become “auxiliary components” for many key technologies, which undoubtedly include additive technologies [4,5]. However, the creation of new polymers, increasing their production, and modifying technological processes for new parameters is a long and resource-intensive process [6,7,8]. A logical solution is the creation of composites based on industrially produced polymers that are capable of processing and have target functional properties. One of the universal functional properties of polymer composite materials is magnetic susceptibility [9,10]. The creation of magnetically susceptible polymeric materials affects all areas of modern materials science, ranging from biomedicine as components of targeted delivery [11], theranostics [12], and destruction of tumors, blood clots, and biovisualization [13,14]. These systems are primarily based on rare earth elements and molecules with a complex topological structure, such as dendrimers [15], hyperbranched polymers [16], and chitosans [17], and are currently of more fundamental interest.

From a practical point of view, in the context of sustainable development and rationalization of resource use, the use of specific polymers is not advisable. In recent years, the trend of imparting specific properties to mass polymer materials has been gaining popularity, such as polyethylene, polypropylene, ABS, and PLA plastic. The use of these polymers is convenient for two reasons. The first is mass production, and as a consequence, the prevalence of technologies for their application. The second, and probably more important aspect, is the possibility of their recycling and reuse.

Thus, in the work of Dimiev et al. [18], a one-stage synthesis of polypropylene/perovskite composites is shown, with subsequent use as filaments for 3D printing. In this work, the entire synthesis of the composite was carried out in the extrusion process, which is more environmentally friendly compared to the classic introduction of a filler. Another study by Adrianna Kania et al. is devoted to the production of magnetically anisotropic composite materials [19]. Roland K. Chen et al. [20] used a similar approach for obtaining magnets of complex shapes, but using photoelectron polymerization. In addition, magnetically soft robots controlled by a magnetic field have received significant interest. For example, in the work of Jiyun Kim, programmable magnetic robots are presented that are capable of taking various complex geometric shapes [21]. The effect of magnetic nanoparticles, both in a magnetic field and without it, on the main functional properties of the polymer matrix is also considered [10].

All these works have in common the fact that the magnetic component of the material is iron oxides and ferrites. On the one hand, this is convenient, since magnetic materials based on them are well-studied and predictable. However, iron oxides and their derivatives are hydrophilic materials, while all mass polymers for additive technologies, with the exception of PLA, are hydrophobic environments. Because of this, it is necessary to introduce additional components into the polymer matrix, which worsens the physical and mechanical properties of the resulting composite.

On the other hand, there are alternative magnetic fillers based on rare earth metals, manganese, cobalt, and cobalt borides. Moreover, they can be divided into two main groups. The first and most common are magnetic materials based on iron with the replacement of one of the iron atoms with cobalt [22], manganese [23], lanthanum and lithium [24], gadolinium [25], and neodymium [26].

However, recent works by Wei Yang et al. [27] and Barbara Albert [22] have shown great interest and the unique magnetic, physical, and chemical properties of individual Fe and Co borides. [28] At the same time, cobalt borides are more thermally stable, which is important for obtaining composites and filaments in the extrusion process. This is well described in the work of Mihael Brunčko et al., devoted to the production of magnetic filaments based on polyamide 12 and thermoplastic polyurethane with the addition of Nd-Fe-B powder [29]. However, in this work, the authors encountered a significant decrease in magnetic characteristics due to the temperature degradation of the magnetic powder. That is why the goal of this work was to develop a new method for synthesizing magnetic cobalt boride nanoparticles whose magnetic properties are resistant to temperature effects. Based on industrially produced thermoplastic, composite materials were obtained. The magnetic and mechanical properties of composites obtained by injection molding and 3D printing were also evaluated.

## 2. Materials and Methods

### 2.1. Materials

CoSO_4_ × 6H_2_O (99%, Alfa Aesar, Haverhill, MA, USA) as a cobalt precursor, polyethylene glycol 4000 (PEG 4000, Sigma-Aldrich, St. Louis, MO, USA, CAS: 25322-68-3, average Mr = 4000 g·mol^−1^, hydroxyl number 28 mg KOH·mg·g^−1^, sodium borohydride (NaBH_4_ 98% Alfa Aesar Cas No: 16940-66-2), native ABS from JSC «Plastic»—Uzlovaya Chemical Plant (Uzlovaya, Russia).

### 2.2. Synthetic Procedures

The composite [PEG/CoB-NP] was synthesized by chemical reduction in an anhydrous medium. CoSO_4_×6H_2_O (m = 25 g) was added to the PEG-4000 melt (m = 100 g, m(KOH) = 23 mg·g^−1^) at 85 °C, after which it was stirred for 120 min, the overall ratio ν(-OH)/ν(Co^2+^) = 1/10 [30,31]. In this form, the nanoparticle precursor can be stored and used as needed.

After the pre-organization stage, the melt was heated to 100 °C and, while stirring with an overhead stirrer, 5 g of NaOH was added to increase the reducing potential, and 50 g of NaBH_4_, over five hours, as a reducing agent. All components were added dry. The melt was stirred until gas evolution ceased; the composite synthesis time for this procedure was 16 h. The resulting composite was removed from the flask and ground in a ball mill with liquid nitrogen cooling. Part of the product was collected for characterization, the remainder was used to obtain the ABS/[PEG/CoB-NP] composite. The [PEG/CoB-NP] composite was loaded into ABS using co-extrusion; the [CoB-NP] content in the polymer matrix was 0.15, 0.3, 0.75, 1.5, 3, 4.5, and 6%.

### 2.3. Composite Materials Production

ABS and nanoparticles were compounded in a LabTech Engineering Scientific LTE (Oldham, UK) 16–40 co-rotating twin screw extruder with a high shear screw configuration (16 mm diameter, L:D—40:1) for better rotation. The screws contain 4 kneading blocks with an inclination angle of 30° and 60°. The processing temperature profile (at feed 190 °C and from the rotation zone to the filler zone 220 °C) was selected in such a way as to provide the appropriate melt viscosity for transfer and at the same time minimize the degradation of both the polymer and fillers. Before processing, the ABS melt was dried at a temperature of 80 °C for 1 h.

During extrusion, the feed rate is maintained at 40 rpm, the melt pressure is 30–35 bar, and the volatile matter degassing is under vacuum illumination at 0.6 bar. The extruded threads are cooled in a water bath and dried in a vacuum. Then the strands are granulated, the granule size is 2.75 mm, the strand feed rate into the granulator is 3–5 m/min, and the resulting granules are re-extruded to obtain a thread. To ensure complete homogenization and uniform distribution of nanoparticles in the final composite, this process was repeated three times. During extrusion, the feed rate is maintained at 40 rpm, the melt pressure is 30–35 bar, and the volatile matter degassing is under vacuum illumination at 0.6 bar. The extruded cooled mixture is placed in a water bath and then dried in a vacuum. The thread diameter of 1.75 mm was regulated by the drawing speed.

#### 2.3.1. Injection Molding

The production of bars for thermomechanical studies was carried out on an RR/TSMP (Ray-Ran, Ray-Ran Test Equipment, Ltd., Nuneaton, UK) injection molding machine in a standard mold for casting in accordance with ISO 178, at a casting temperature of 230 °C, and a mold temperature of 70 °C. The production of 30 mm and 4 mm thick disks and cylinders was carried out under similar conditions.

#### 2.3.2. D Printing of Mechanical Test Samples

An Anycubic Kobra 3D printer (Shenzhen Anycubic Technology Co., Ltd., Shenzhen, China) with a single FDM nozzle was used to produce all mechanical test samples in the form of 60 × 10 × 4 mm bars. The filament was extruded through a 0.6 mm diameter nozzle heated to 245 °C, polypropylene tape was glued to the table surface for better adhesion of the melt, and the table was heated to 40 °C. Print speed was 15 mm/min. All samples were printed with a print layer height of 0.3 mm and 100% material infill.

### 2.4. Methods

#### 2.4.1. DSC

The differential scanning calorimetry was studied using a NETZSCH DSC 214 Polyma instrument (NETZSCH, Graz, Austria). The samples of 7.0 ± 0.5 mg weight were cut from the granules and placed in aluminum crucibles (25 μL) with lids. The samples were heated from 25 to 230 °C, annealed in the melt for 3 min, cooled down to 50 °C again, and finally melted by bringing them to 230 °C. All heating and cooling ramps were performed at a rate of 10 °C·min^−1^ and a nitrogen flow rate of 60 mL·min^−1^.

Melt flow measurements were carried out in standard mode for ABS according to ISO 19062-2. The measurements were carried out at a temperature of 220 °C with a load of 10 kg. The measurement used a standard capillary with a nominal length of 8000 mm and a nominal internal diameter of 2095 mm. The sample preheating time was 7 min.

#### 2.4.2. TG-DTG

TGA analysis was performed in a NETZSCH TG 209 F1 Libra analyzer under an air atmosphere (NETZSCH, Graz, Austria). Temperature was elevated from 30 to 900 °C at 10 °C min^−1^. The sample mass was about 50 mg. The experiments were performed twice for reproducibility. The reproducibility was good with a standard error of ±0.5 °C for the same conversion rate. Prior to measurements, the calibration of the thermocouple was performed.

#### 2.4.3. Fourier-Transform Infrared Spectroscopy

The FT-IR spectra were recorded on a Spectrum 400 (PerkinElmer, Waltham, MA, USA) ATR-FT-IR spectrometer. The FT-IR spectra from 4000 to 400 cm^−1^ were considered in this analysis. The spectra were measured with 1 cm^−1^ resolution and 32-scan coaddition.

#### 2.4.4. DMA

Dynamic mechanical measurements of composite specimens with a geometry of 60 × 10 × 4 mm were carried out using a DMA 242 analyzer (NETZSCH, Graz, Austria) in the double-arm bending mode with a span length of 32 mm. The specimens were tested at temperatures from 30 to 140 °C at a heating rate of 5 °C·min^−1^ with a frequency of 1 Hz under a dynamic load of 5 N. Each measurement was repeated at least three times.

#### 2.4.5. Nanoparticle Tracing Analyzer

The colloidal properties were studied by the NTA method on a NanoSight LM—10 instrument (Malvern Panalytical, Malvern, UK). CMOS cameras C11440-50B with an FL-280 Hamamatsu Photonics (Shizuoka, Japan) image capture sensor were used as a detector. Measurements were taken in a special cuvette for aqueous solutions, equipped with a 405 nm laser (version CD, S/N 2990491) and a sealing ring made of Kalrez material. The temperature was taken with an OMEGA HH804 contact thermometer (Engineering, Inc., Stamford, CT, USA) for all measurements. Samples for analysis were detected and injected into the measuring cell with a 1 mL glass 2-piece syringe (tuberculin) through the Luer (Hamilton Company, Reno, NV, USA). To increase the statistical dose, the sample was pumped through the measuring chamber by using a piezoelectric dispenser. Each sample was detected sequentially six times; the recording time was sequential and amounted to 60 s. For processing the footage of the Nanosight instrument, NTA 2.3 software applications (build 0033) and OriginPro program package were used; the Gaussian function was used throughout, as described previously [16,32,33,34].

#### 2.4.6. Powder X-Ray Diffraction

The XRD was acquired with a Bruker D8 Advance (Billerica, MA, USA) with Cu Kα irradiation (λ = 1.5418 Å) in the Bragg-Brentano geometry; the rate was 0.18 °/min; the range of 2θ angles was from 7° through 100°; the step was 0.015°.

#### 2.4.7. Microscopy

Transmission electron microscopy (TEM) imaging was carried out with a Hitachi HT7700 Excellence transmission electron microscope (Hitachi High-Technologies Corporation, Tokyo, Japan) at an accelerating voltage of 100 kV in the TEM mode.

The scanning electron microscopy (SEM) images were acquired with the field-emission high-resolution scanning electron microscope Merlin from Carl Zeiss (Jena, Germany) at an accelerating voltage of incident electrons of 5 kV and a current probe of 300 pA.

#### 2.4.8. Magnetic Properties

The magnetic properties were measured using a PPMS-9 instrument (Quantum Design USA, San Diego, CA, USA) equipped with a sample vibration magnetometer. ZFC and FC measurements were performed in a 100 Oe magnetic field. The field dependences of the magnetization were measured at 5–300 K in the magnetic field range from −1T to 1T.

#### 2.4.9. Dielectric Measurements

The permittivity and conductivity values were calculated from the impedance, measured with the Novocontrol BDS Concept-80 impedance analyzer (Novocontrol Technologies, Montabaur, Germany), supplied with the automatic temperature control provided by the cryosystem QUATRO (the temperature uncertainty is ±0.5 °C) (Novocontrol Technologies, Montabaur, Germany). The samples were circle-shaped with a diameter of 30.0 mm and a thickness of 3.0 mm. A sample was placed between the two gold-plated electrodes of the capacitor. The capacitor was attached to the thermostated testing head. The measurements were conducted in the frequency range of 0.1 Hz to 10 MHz.

## 3. Results

### 3.1. [PEG/CoB-NP] Nanocomposite

The synthesis of nanoparticles took place in a polymer melt, without the use of solvents, which is one of the key modern trends in materials chemistry. In general, the production of cobalt boride can be represented as:
(1)
$$2{CoSO}_{4}+4{NaBH}_{4}+{18H}_{2}O\overset{NaOH\;(85\;C^{0})}{\to }2CoB+{2Na}_{2}{SO}_{4}+{23H}_{2}+{2B\left(OH\right)}_{3}$$


According to TEM data, the resulting composite is spheroid nanoparticles with an average size of 24 ± 3 nm in diameter (Figure 1A,B). Based on SAED data (Figure 1A,B inset), the nanoparticles are cobalt boride with a cubic lattice, with CoB (210) and CoB (212) planes.

The resulting nanoparticles are highly soluble in water, which can expand their applications in biomedicine. According to NTA data, at a concentration of 1 mg·mL^−1^, nanoparticles with diameters of 50, 72, 100, 125, etc., i.e., multiples of 25 nm, are present in the solution (Figure 1C). This indicates the tendency of CoB nanoparticles, with PEG 4000 as a stabilizer, to aggregate. The average particle size according to NTA is 102 ± 8 nm, and the particle concentration is 4.96 ± 0.66 × 10^8^ particles·mL^−1^. The analysis by FT-IR spectroscopy showed that not only cobalt borides but also cobalt oxides are formed. The FT-IR spectra show vibrations of the B-O and B-H bonds, resulting in a broadening of the bands at 1400 cm^−1^ (Figure 1D,E). The appearance of cobalt oxides is probably due to insufficient chelating activity and high conformational mobility of PEG 4000. We observed a similar effect in our previous work. [32] Comparison of the infrared spectra with native CoB clearly shows the appearance of spectral lines of the Co-B bond in the resulting composite.

The XRD data are consistent with both the SAED and FT-IR spectra (Figure 1F). The [PEG/CoB-NP] composite exhibits clear cobalt boride reflections and Co_3_O_4_ reflections detected in the infrared spectra. Thus, the structure of the obtained nanoparticles is a core-shell structure, where cobalt boride acts as a core, and cobalt oxides act as a shell. This will allow us to further apply this nanoparticle synthesis technology to obtain conjugates, since individual cobalt borides are extremely inert. To summarize, in this part of the work, a new low-temperature method for obtaining monodisperse particles of cobalt boride nanoparticles without using a solvent was developed.

### 3.2. ABS/[PEG/CoB-NP] Composite

The next stage of this work was the introduction of the obtained nanoparticles into the polymer matrix. ABS plastic was chosen as a polymer matrix for several reasons. Firstly, sufficient “hydrophilicity”; secondly, prevalence and availability; and thirdly, wide application in 3D printing.

The first question that needs to be answered is whether the performance parameters of the polymer matrix have worsened after the introduction of composite materials into it. The key characteristic of any thermoplastic has been and will be resistance to thermal-oxidative degradation. This is especially important since it is impossible to maintain the inertness of the atmosphere in the production of polymer products or in-home 3D printing conditions.

Carrying out thermogravimetric analysis in an atmosphere of synthetic air showed that native ABS burns without residue at 600 °C, in the case of ABS/[PEG/CoB-NP] composite, a non-combustible residue is observed, 10% of the original, which is Co_3_O_4_ (Figure 1A). The most important result, in our opinion, is the increase in the temperature of thermal-oxidative destruction by 30 °C for the ABS/[PEG/CoB-NP] composite compared to naive ABS plastic.

In addition to increasing the oxidation stability during heating, the introduction of [PEG/CoB-NP] nanoparticles turned out to allow for quite precise control of the glass transition temperature. DSC studies of ABS/[PEG/CoB-NP] composites containing different amounts of nanoparticles show a stable increase in the glass transition temperature from 107.5 °C to 112 °C, depending on the nanoparticle loading (Figure 2B). At the same time, based on the values of the phase transition temperature, the use of the obtained [PEG/CoB-NP] nanoparticles allows for controlling the glass transition temperature of ABS/[PEG/CoB-NP] by up to 0.5 °C. Also, with an increase in the content of nanoparticles in the ABS/[PEG/CoB-NP] composite, an increase in the thermal conductivity of the composites is observed (Figure 2C).

The glass transition temperature of the samples increases possibly due to the increase in the specific heat capacity and thermal conductivity of the composites when introducing nanoparticles. Nanoparticles absorb part of the heat and dissipate it, which can lead to an increase in the glass transition temperature of the composites with increasing concentration.

The next stage of this work was to study the physical and thermomechanical properties of the obtained composite materials. In this case, a comparison was made between standard samples obtained by 3D printing and those obtained by injection molding.

The first thing we encountered was the inability to obtain filaments from native ABS, while obtaining samples by injection molding was not a problem. Therefore, an experiment was conducted to evaluate the fluidity index and melt flow rate; the results are presented in Table 1.

Table 1 shows that native ABS, which was used in this work, has the lowest MFI values, which explains the impossibility of obtaining a filament. In the case of ABS/[PEG/CoB-NP] composite materials, a division into two groups is observed. In the range of [PEG/CoB-NP] content from 0.15 to 0.75%, the MFI values remain virtually unchanged, after which they increase sharply, reaching an average MFI value of 20 cm^3^ × 10 min^−1^. These values compare to the reference ABS ESUN. Such a sharp jump in viscosity values, as we believe, is associated with an increase in the number of magnetic and adhesive interactions of filler particles, due to their approach to each other with increasing concentration in the composite.

This conclusion is based on the fact that at a filler concentration above 1.5%, cobalt boride signals are observed in the XRD spectra of composites (Figure 3). That is, in the polymer matrix, the ordering of the filler increases at a concentration above 1.5%, due to which the intensity of the cobalt boride reflexes increases.

A logical continuation was the study of the thermomechanical properties of the obtained composites. Figure 4 shows the curves of the dynamic elastic modulus versus temperature for composite samples obtained by casting (Figure 4A), printed on a 3D printer (Figure 4B), and a comparison of the maximum elastic modulus (Figure 4C).

The difference in the values of the dynamic modulus of elasticity of commercial ABS, compared to the obtained composites, in the range from 1 to 3% of the loading of nanoparticles was 15 ± 2%, regardless of the method of obtaining the samples. First of all, this is due to the thermal-oxidative degradation and the decrease in the length of the polymer chains during the production of the composite material, as described in Part 2. However, as can be seen from Figure 4, the addition of [PEG/CoB-NP], in the range from 0.15 to 3%, does not significantly affect the difference in the thermomechanical properties of the composite.

Regardless of whether it is cast or 3D printed, the E′ values vary within a narrow range. The glass transition temperature in the same range is 113 ± 5 °C for samples obtained by injection molding, and 118 ± 5 °C obtained by 3D printing. A further increase in the concentration of nanoparticles in the polymer leads to a significant decrease in the elastic modulus, which we associate with two factors: an increase in the concentration of PEG, which is a plasticizer, and an increase in the viscosity of the composite due to the physical and magnetic interaction of the filler particles.

The obtained composite samples were examined by SEM (Figure 5).

As can be seen from Figure 5, in the range of [PEG/CoB-NP] concentrations from 0.15 to 1.5%, no agglomeration of nanoparticles occurs; the main fraction of particles is the fraction of 28 ± 3 nm in diameter (Figure 5A–D). A further increase in the concentration of the nanocomposite filler leads to an increase in the size of aggregates to 90 ± 10 nm (Figure 5E–H). Probably, the deterioration of the thermomechanical properties observed at high loading (Figure 4C) is associated with the lack of the possibility of uniform distribution of these particles in the polymer matrix due to their size.

### 3.3. Magnetic and Dielectric Properties ABS/[PEG/CoB-NP] Composite

The main objective of this work was to impart magnetic properties to an industrially produced thermoplastic. First, the bulk magnetic susceptibility of all the obtained composites was tested (Figure 6B), and of course, for simple interaction with a magnet (Figure 6A), the weight of the samples was 10 ± 1 g. As can be seen from the curve of the bulk magnetic susceptibility, in the range of nanoparticle concentration from 0.15 to 4%, the magnetic susceptibility increases linearly, reaching a maximum of 1.3 × 10^5^ 
$\chi_{V}^{SI}$
, and reaching the limit at a filler concentration of 6%. The next study was to study the type of magnetism and the magnetic diameter of the obtained nanoparticles, similar to our previous work [35]. The magnetic diameter was estimated specifically in the final composite since several heating and cooling cycles will inevitably lead to a change in the nanoparticles.

Based on the field and temperature dependence of the sample magnetization (Figure 6), the magnetic state of the resulting composite ensemble can be defined as superparamagnetic. The blocking temperature for the ABS + 6% [PEG/CoB-NP] sample was 13 K

Therefore, the approach to assessing the magnetic diameter of nanoparticles in the final composite is of not only scientific but also practical interest to the industry. Comparison of the magnetic diameter of particles in the composite obtained by magnetometry with the data obtained by TEM, SEM, ASM, DLS, and NTA allows us to unambiguously prove whether particle agglomeration occurred in the final composite. In our case, the magnetic diameter of nanoparticles in the most loaded ABS/[PEG/CoB-NP] composite was 25 nm, which is completely consistent with the TEM data of the original PEG/CoB-NP nanoparticles (Figure 1). This proves that despite the agglomeration of the particles themselves into larger aggregates at high loads, the magnetite cores of CoB nanoparticles remain individual.

The final stage of this study was the evaluation of the dielectric constant of the obtained composites. First, we were afraid that the introduction of a large number of magnetic particles would worsen the electrical insulation properties of ABS. As can be seen from Figure 7, the total dielectric constant of the composite material remained at the level of the original polymer. In the frequency range from 10^0^ to 10^6^ Hz, with an increase in the concentration of nanoparticles, a linear increase in dielectric constant is observed.

In the case of dielectric parameters of composite materials, the properties of the composite also change significantly when the concentration reaches 1.5%. In the loading range of 0.15–1.5%, an increase in the efficiency of charge transfer is observed, which is logical and consistent with the Maxwell–Wagner effect. However, a further increase in the CoB material loading in the range from 3 to 6% leads to a sharp shift in the absorption maximum to the ultra-low frequency region, which is probably due to agglomeration and complication of charge transfer from the surface of particles. This confirms our assumption about an increase in the viscosity of the melt flow due to magnetic interactions between particles and SEM images (Figure 5).

## 4. Conclusions

A new and low-temperature approach to obtaining cobalt boride nanoparticles with a diameter of 25 nm has been developed. The resulting material has low polydispersity, is soluble in water, and is synthesized from available reagents. Using the obtained nanoparticles as additives to native ABS, which is incapable of forming a filament, allows obtaining a polymer composite filament with magnetic susceptibility.

By varying the loading of [PEG/CoB-NP] in the polymer, it is possible to finely control the glass transition temperature in increments of 0.5 °C, which expands the possibilities of its application. Imparting magnetic properties in combination with a shift in the maximum permittivity to the short-wave region will expand the scope of ABS as an electrical material.

Despite several processing cycles, ABS/[PEG/CoB-NP] nanoparticles in the final composite remain individual at a loading of 0.15 to 1.5%. The particle diameter is 28 ± 3 nm, which coincides with the diameter of individual particles obtained by TEM. Increasing the concentration in the range from 3% leads to coarsening of nanoparticles into aggregates with a diameter of 90 ± 10 nm. This effect leads to a decrease in thermomechanical properties. However, the calculation of the magnetic diameter using the Langevin function shows that the magnetic cores of the nanoparticles remain unchanged and their diameter is still 25 nm. We assume that this is precisely why charge transfer from the surface of nanoparticles becomes significantly more complicated, and as a consequence, the maximum absorption of the permittivity shifts.

## Figures and Tables

**Figure 1 polymers-17-00038-f001:**
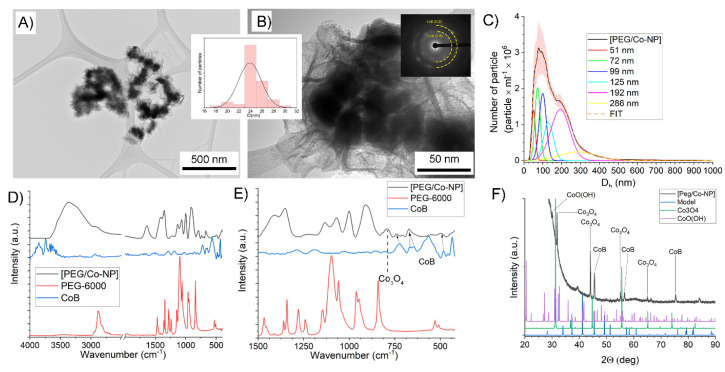
TEM images with size distribution and SAED of [PEG/CoB-NP] nanocomposite (**A**,**B**); Nanoparticle distribution in aqueous solution by NTA method (**C**); FT-IR spectra of [PEG/CoB-NP] nanocomposite (**D**,**E**); XRD spectra of [PEG/CoB-NP] nanocomposite (**F**).

**Figure 2 polymers-17-00038-f002:**
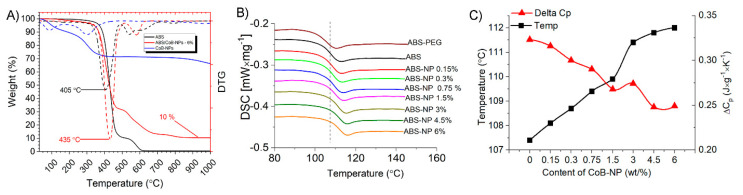
TG-DTG curves of native ABS, [PEG/CoB-NP] composite and ABS/[PEG/CoB-NP] filaments with 6% loading [PEG/CoB-NP] (**A**); DSC curves of ABS/[PEG/CoB-NP] composite with different loading of nanoparticles (**B**); heat capacity of phase transition ABS/[PEG/CoB-NP] composite with different loading of nanoparticles (**C**).

**Figure 3 polymers-17-00038-f003:**
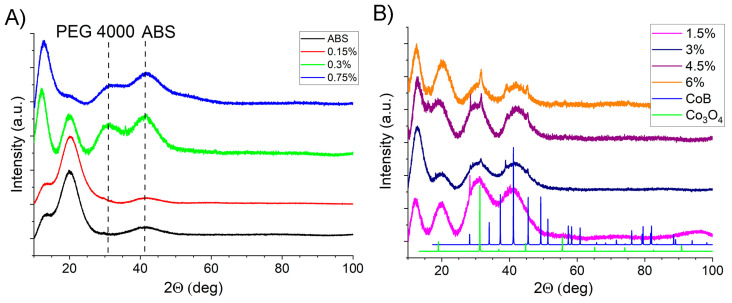
XRD spectra of filaments: ABS granules, ABS + 0.15% [PEG/CoB-NP], ABS + 0.3% [PEG/CoB-NP], ABS + 0.75% [PEG/CoB-NP] (**A**); ABS + 1.5% [PEG/CoB-NP], ABS + 3% [PEG/CoB-NP], ABS + 4.5% [PEG/CoB-NP], ABS + 6% [PEG/CoB-NP (**B**).

**Figure 4 polymers-17-00038-f004:**
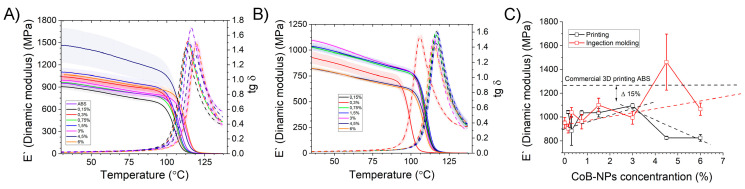
Thermomechanical properties ABS and ABS/[PEG/CoB-NP] composite with different loading of nanoparticles, by the DMA method: produced by injection molding (**A**); produced by 3D printing (**B**); comparison of the obtained samples by thermomechanical properties (**C**).

**Figure 5 polymers-17-00038-f005:**
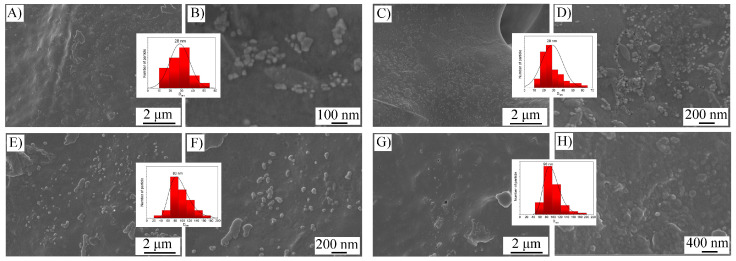
SEM images of ABS/[PEG/CoB-NP] filament composites at different magnifications with particle size distribution: ABS + 0.3% [PEG/CoB-NP] (**A**,**B**); ABS + 1.5% [PEG/CoB-NP] (**C**,**D**); ABS + 3% [PEG/CoB-NP] (**E**,**F**); ABS + 6% [PEG/CoB-NP (**G**,**H**).

**Figure 6 polymers-17-00038-f006:**
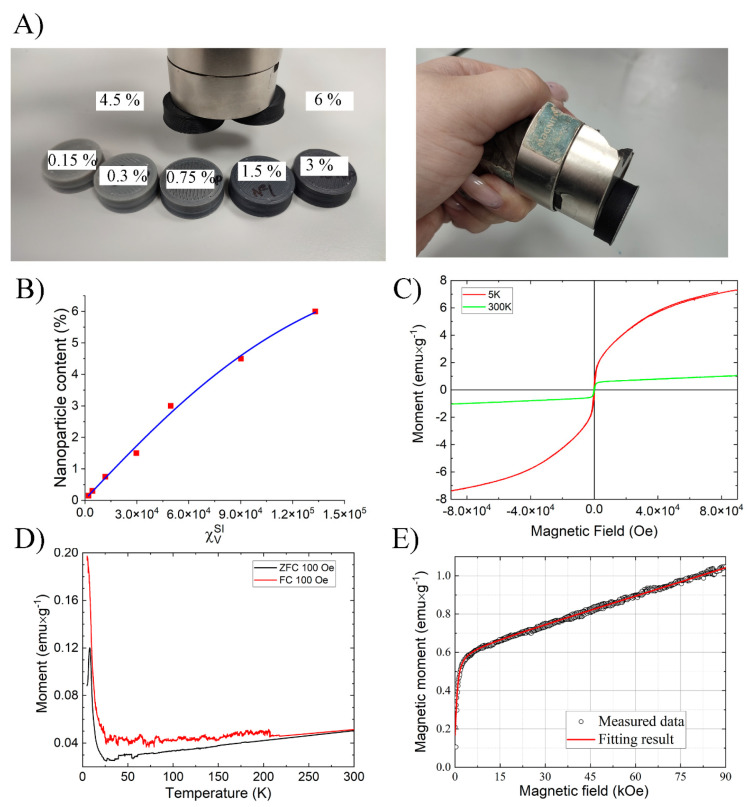
Magnetic properties of the composite materials: Interaction of samples obtained by 3D printing with a neodymium magnet (sample mass 10 g) (**A**); Magnetic susceptibility of samples obtained by 3D printing as a function of nanoparticle loading (**B**); Magnetic hysteresis for ABS + 6% [PEG/CoB-NP] measured at 5 and 300K (**C**); ZFC–FC curves for ABS + 6% [PEG/CoB-NP] (**D**); Determination of the magnetic diameter of the resulting nanoparticles (**E**).

**Figure 7 polymers-17-00038-f007:**
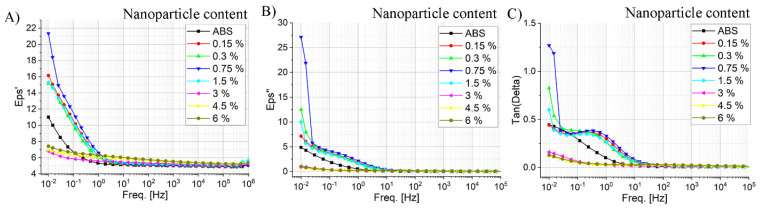
Dielectric constant curves of the ABS/[PEG/CoB-NP] composite at different loadings of nanoparticles obtained by 3D printing: dependence of ε′ on frequency range from 10^−2^ to 10^6^ Hz (**A**); dependence of ε″ on frequency range from 10^−2^ to 10^6^ Hz (**B**); dielectric loss tangent on frequency range from 10^−2^ to 10^6^ Hz (**C**).

**Table 1 polymers-17-00038-t001:** Melt Flow Index (MFI) and Melt Flow Rate (MFR) for native ABS, ABS ESUN comparison standard, and ABS/[PEG/CoB-NP] filament composite with different nanoparticle loading.

Sample	MFI, cm^3^ × 10 min^−1^	MFR, g × 10 min^−1^
Native ABS	12.00	12.60
ABS ESUN	19.55	20.53
ABS + 0.15% [PEG/CoB-NP]	12.47	13.09
ABS + 0.3% [PEG/CoB-NP]	13.12	13.78
ABS + 0.75% [PEG/CoB-NP]	13.64	14.32
ABS + 1.5% [PEG/CoB-NP]	22.32	24.48
ABS + 3% [PEG/CoB-NP]	18.12	19.02
ABS + 4.5% [PEG/CoB-NP]	20.44	21.46
ABS + 6% [PEG/CoB-NP]	19.57	20.55

## Data Availability

The data presented in this study are available in the Appendix A.

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
