# Peer review of "Control of Dielectric, Mechanical, and Thermal Properties of a Polymer Composite Based on ABS Using CoB Nanoparticles"

_polymers, 2024, doi:10.3390/polym17010038_

Round 1

Reviewer 1 Report

Comments and Suggestions for Authors

For the comments and suggestions for authors see the attached PDF file

Author Response

 Thank you very much for taking the time to review this manuscript. Thank you and your comments concerning our manuscript entitled “Control of dielectric, mechanical and thermal properties of a polymer composite based on ABS use CoB nanoparticles” (ID: polymers-3378528). Those comments are all valuable and very helpful for revising and improving our manuscript, as well as the important guiding significance to our researches. We have studied comments carefully and have made correction which we hope meet with approval. Please find the detailed responses below and the corresponding revisions/corrections highlighted/in track changes in the re-submitted files.

Point-by-point response to Comments and Suggestions for Authors

  1. Authors should spend more time and effort editing and proofreading their manuscript a.

There are several instances of typos and/or parts of sentences missing i.e in lines 141, 202, 361

  • The description of the DSC method has been completely rewritten in connection with the comment from point 3..

Response:

The differential scanning calorimetry was studied using a NETZSCH DSC 214 Polyma instrument.  The samples of 7.0 ± 0.5 mg weight were cut from the granules and placed in aluminum crucibles (25 μl) with lids.The samples were heated from 25 to 230 °C, annealed in the melt for 3 min, cooled down to 50 °C again, and finally melted by bringing them to 230 °C. All heating and cooling ramps were performed at a rate of 10 °C/min and a nitrogen flow rate of 60 ml/min

  • he permittivity and conductivity values were calculated from the impedance corrected to:

Response:

The permittivity and conductivity values were calculated from the impedance

  • . In the frequency range from 100 to 106, with an increase corrected to: .

Response:

In the frequency range from 100 to 106 Hz, with an increase

  1. Figure captions should follow the same format throughout the manuscript.

Response:

All figures and captions are brought to a uniform form.

  1. The caption for figure 5 needs to be corrected.

Response:

All caption are corrected

  1. Figure 6 A,B depicts the dependance of ε’ and ε’’ on frequency or all nanocomposites and not on the concentration as stated In the caption for Figure 6 (A,B)

Response:

Caption figure 7 (old figure 6) correctrd to :

Figure 7. Dielectric constant curves of the ABS / [PEG/CoB-NP] composite at different loading of nanoparticles: A) dependence of ε` from frequency range from 10-2 to 106 Hz; B) dependence of ε`` from frequency range from 10-2 to 106 Hz; ; C) dielectric loss tangent from frequency range from 10-2 to 106 Hz

  1. It’s sometimes confusing to distinguish which fabrication method was used for the preparation of the composite specimens for each employed characterization technique.

Response:

Until section 3.2, only the nanoparticles themselves were discussed. In section 3.2, ABS/[PEG/CoB-NP] nanoparticle composites were studied. The different methods of sample preparation were discussed only in the context of Figure 4.

For better understanding of which sample is being studied, the corresponding changes have been added to the figure captions.

  1. In Materials and Methods is stated that the degree of crystallinity was calculated using the “heat of melting of fully crystalline polypropylene”. If the statement is not just a copy/paste from a previous published paper, a justification is needed. Also, there is absolutely no mention of crystallinity in the results section.

Response:

We thank the reviewer for the comment, in paragraph 2.4.1. I really copy/paste from the previous publication. The error has been corrected, the mention of the “degree of crystallinity” has been excluded from the manuscript.

  1. Authors spend a reasonable part of the manuscript, and rightfully so, discussing the morphology and particle size of [PEG/CoB-NP]. There is no mention of the dispersion and distribution of the nanoparticles in the employed ABS matrix even though the aforementioned parameters influence the properties of composite systems. TEM or SEM images of the nanocomposite systems should be added to the revised version of the manuscript.

Response:

We thank the reviewer for this important comment. We performed additional experiments as suggested by the reviewer. In the Methods section, the following is added:

The scanning electron microscopy (SEM) images were acquired with the field-emission high-resolution scanning electron microscope Merlin from Carl Zeiss (Jena, Germany) at an accelerating voltage of incident electrons of 5 kV and a current probe of 300 pA.

Section 3.2 has been supplemented with SEM images of cobalt boride nanoparticle size determination.

  1. It is stated in the conclusions section that “we clearly prove that even in the most highly loaded materials, agglomeration and coarsening of nanoparticles does not occur during extrusion”. However, right above in the discussion of the dielectric properties, authors state “…which is probably due to agglomeration and complication of charge transfer from the surface of particles .”This contradiction can only be resolved by providing SEM or TEM images of the nanocomposites.

6.There is absolutely no discussion over the mechanical properties of the nanocomposites. Authors should provide a scientifically sound commentary regarding the DMA results or discard the entire section.

Figure 4 has been edited. Thermomechanical properties for the samples obtained by injection molding from native ABS have been added. A more detailed discussion of them has also been added to the manuscript.

Response:

The difference in the values of the dynamic modulus of elasticity of commercial ABS, compared to the obtained composites, in the range from 1 to 3% of the loading of nanoparticles was 15±2%, regardless of the method of obtaining the samples. First of all, this is due to the thermal-oxidative degradation and the decrease in the length of the polymer chains during the production of the composite material, as described in Part 2. However, as can be seen from Figure 4, the addition of [PEG / CoB-NP], in the range from 0.15 to 3%, does not significantly affect the difference in the thermomechanical properties of the composite.

Regardless of whether it is cast or 3D printed, the E` values vary within a narrow range. The glass transition temperature in the same range is 113±5 °C for samples obtained by injection molding, and 118±5 °C obtained by 3D printing. A further increase in the concentration of nanoparticles in the polymer leads to a significant decrease in the elastic modulus, which we associate with two factors: an increase in the concentration of PEG, which is a plasticizer, and an increase in the viscosity of the composite due to the physical and magnetic interaction of the filler particles.

  1. Regarding the magnetic properties. “The next study was to study the type of magnetism, determine the Curie-Weiss temperature and the magnetic diameter of the obtained nanoparticles, similar to our previous work”

Authors state:

And

  1. “The next study was to study the type of magnetism, determine the Curie-Weiss temperature and the magnetic diameter of the obtained nanoparticles, similar to our previous work”

And

  1. “Based on the field dependence of the sample magnetization (Figures 5C and 5E), the magnetic state of the resulting composite ensemble can be defined as superparamagnetic, while individually within each nanoparticle, antiferromagnetic, ferrimagnetic or ferromagnetic ordering of magnetic moments can be realized. Based on the temperature dependence of magnetization, it can be assumed that all of the above scenarios are realized, since it is impossible to unambiguously describe this dependence within the framework of one model”

How can anyone determine the Curie-Weiss temperature in a system that exhibits superparamagnetic behaviour?

Response:

Answer: We agree with this remark. The text of the article has been amended..

The next study was to study the type of magnetism , determine the Curie-Weiss temperature and the magnetic diameter of the obtained nanoparticles, similar to our previous work [35].

Replaced by

The next study was to study the type of magnetism and the magnetic diameter of the obtained nanoparticles, similar to our previous work [35].

Furthermore, regarding the temperature dependance of magnetization, authors should provide sufficient experimental data that could imply their claim about any antiferromagnetic, ferrimagnetic or ferromagnetic ordering within each nanoparticle. I am personally very skeptical about how this statement goes along with their assessment of superparamagnetic behaviour of the systems.

Response:

The point is that we only state the superparamagnetic state of the entire ABS/PEG-CoB-NP composite material sample. Any magnetic ordering can be realized inside the particles themselves and/or their agglomerates, which leads to the particle simply forming its own resulting magnetic moment, which allows them to be interpreted as superparamagnetic.

We cannot say for sure the nature of the formation of this moment and cannot guarantee that it is the same for all particles; this is a separate area of research that we plan to pursue in the future.

In the context of this work, we wanted to say that:

Inside each particle, the spins are ordered in one way or another by an undefined mechanism, thereby forming the resulting moment of this particle. Inside each particle, this happens according to its own scenario - it may be the same for all particles, or it may be strictly individual. We are only interested in the result of these processes, which leads to the fact that each particle has its own resulting magnetic moment, which makes this particle superparamagnetic. Next, we use the Langevin model to describe and characterize this ensemble as a whole, rather than individually, and the description of them as superparamagnetic particles has the greatest correlation. The resulting magnetic moment is then effectively the average magnetization of each particle.

Finally, they should provide at least the blocking temperature value.

Response:

The blocking temperature is 13 degrees K. The corresponding changes have been made to the article.

Response:

The conclusions have been changed and supplemented according to new data obtained based on the comments and suggestions of the reviewer:

Despite several processing cycles, ABS/[PEG/CoB-NP] nanoparticles in the final composite remain individual at a loading of 0.15 to 1.5%. The particle diameter is 28±3 nm, which coincides with the diameter of individual particles obtained by TEM. Increasing the concentration in the range from 3% leads to coarsening of nanoparticles into aggregates with a diameter of 90±10 nm. This effect leads to a decrease in thermomechanical properties. However, calculation of the magnetic diameter using the Longevin function shows that the magnetic cores of the nanoparticles remain unchanged and their diameter is still 25 nm. We assume that this is precisely why charge transfer from the surface of nanoparticles becomes significantly more complicated, and as a consequence, the maximum absorption of the permittivity shifts.

Thank you very much for taking the time to review this manuscript. Thank you and your comments concerning our manuscript entitled “Control of dielectric, mechanical and thermal properties of a polymer composite based on ABS use CoB nanoparticles” (ID: polymers-3378528). Those comments are all valuable and very helpful for revising and improving our manuscript, as well as the important guiding significance to our researches. We have studied comments carefully and have made correction which we hope meet with approval. Please find the detailed responses below and the corresponding revisions/corrections highlighted/in track changes in the re-submitted files.

  1. In the Page 5 line 215, the specific conditions or corresponding sources of the chemical reaction equations should be given.

Response:

The following changes have been made to the manuscript:

  1. what is the different in Fig.1 (D and E)?

Response:

Figure E is a narrow region of the FT-IR spectrum in the range from 1500 to 450 cm-1. In this region, characteristic vibrations of the Co-B bond are present. Unfortunately, due to the low dipole moment, these signals have low intensity and are weakly visible in the overall spectrum. Therefore, to prove the presence of cobalt borides in the PEG/CoB-NP composite, we presented a separate region of the spectrum where they are visible.

  1. Some of the figures are too small, and the text in the figures are not clear, all of the figures are suggested to revised.

Response:

Thank you for your comment. In the new version of the manuscript, all the figures are inserted with a resolution of 600 dpi. Since text editors tend to compress images, as additional manuscript files, all the figures are duplicated in attachment files.

  1. Why the glass transition temperature of ABS/[PEG/CoB-NP] composites only change from 107.5 °C to 112 °C should be further analysis?

Response:

The following text has been added to the text of the manuscript, in section 3.2:

The glass transition temperature of the samples increases possibly due to the increase in the specific heat capacity and thermal conductivity of the composites when introducing nanoparticles. Nanoparticles absorb part of the heat and dissipate it, which can lead to an increase in the glass transition temperature of the composites with increasing concentration.

  1. The XRD spectra in Figure.3 should label each diffraction peak.

Response:

Figure 3 is modified. Panel A) shows the peaks of ABS and PEG4000. Panel B) shows the calculated diffraction peaks of cobalt boride and oxide.

  1. The in-depth mechanism of electromagnetic properties change should be elaborated.

Response:

The main objective of this work is to show the possibilities of using cobalt nanocomposites as additives for thermoplastics and to prove that they retain their specific properties after several processing cycles. The study of the electromagnetic properties of such systems is a separate and interesting task. It is necessary to conduct studies at different temperatures and to involve the Mie scattering theory. Research in this direction is planned as a further development and expansion of this work.

Reviewer 2 Report

Comments and Suggestions for Authors

1.  In the Page 5 line 215, the specific conditions or corresponding sources of the chemical reaction equations should be given.

2. what is the different in Fig.1 (D and E)?

3. Some of the figures are too small, and the text in the figures are not clear, all of the figures are suggested to revised.

4. Why the glass transition temperature of ABS/[PEG/CoB-NP] composites only change from 107.5 °C to 112 °C should be further analysis?

5. The XRD spectra in Figure.3 should label each diffraction peak.

6. The in-depth mechanism of electromagnetic properties change should be elaborated.

Author Response

Thank you very much for taking the time to review this manuscript. Thank you and your comments concerning our manuscript entitled “Control of dielectric, mechanical and thermal properties of a polymer composite based on ABS use CoB nanoparticles” (ID: polymers-3378528). Those comments are all valuable and very helpful for revising and improving our manuscript, as well as the important guiding significance to our researches. We have studied comments carefully and have made correction which we hope meet with approval. Please find the detailed responses below and the corresponding revisions/corrections highlighted/in track changes in the re-submitted files.

  1. In the Page 5 line 215, the specific conditions or corresponding sources of the chemical reaction equations should be given.

Response:

Corrected reaction equation in the appendix file

  1. what is the different in Fig.1 (D and E)?

Response:

Figure E is a narrow region of the FT-IR spectrum in the range from 1500 to 450 cm-1. In this region, characteristic vibrations of the Co-B bond are present. Unfortunately, due to the low dipole moment, these signals have low intensity and are weakly visible in the overall spectrum. Therefore, to prove the presence of cobalt borides in the PEG/CoB-NP composite, we presented a separate region of the spectrum where they are visible.

  1. Some of the figures are too small, and the text in the figures are not clear, all of the figures are suggested to revised.

Response:

Thank you for your comment. In the new version of the manuscript, all the figures are inserted with a resolution of 600 dpi. Since text editors tend to compress images, as additional manuscript files, all the figures are duplicated in attachment files.

  1. Why the glass transition temperature of ABS/[PEG/CoB-NP] composites only change from 107.5 °C to 112 °C should be further analysis?

Response:

The following text has been added to the text of the manuscript, in section 3.2:

The glass transition temperature of the samples increases possibly due to the increase in the specific heat capacity and thermal conductivity of the composites when introducing nanoparticles. Nanoparticles absorb part of the heat and dissipate it, which can lead to an increase in the glass transition temperature of the composites with increasing concentration.

  1. The XRD spectra in Figure.3 should label each diffraction peak.

Response:

Figure 3 is modified. Panel A) shows the peaks of ABS and PEG4000. Panel B) shows the calculated diffraction peaks of cobalt boride and oxide.

  1. The in-depth mechanism of electromagnetic properties change should be elaborated.

Response:

The main objective of this work is to show the possibilities of using cobalt nanocomposites as additives for thermoplastics and to prove that they retain their specific properties after several processing cycles. The study of the electromagnetic properties of such systems is a separate and interesting task. It is necessary to conduct studies at different temperatures and to involve the Mie scattering theory. Research in this direction is planned as a further development and expansion of this work.

Round 2

Reviewer 1 Report

Comments and Suggestions for Authors

I want to thank the authors for addressing my comments regarding their manuscript. The present work would be suitable for publication after the implementation of two minor alterations.

1. A thorough proofreading in the captions before publication. There are still some typos (Figure 4) and words missing (Figure 6).

2. The following passage 

"Based on the field dependence of the sample magnetization (Figures 6C and 5E), the magnetic state of the resulting composite ensemble can be defined as superparamagnetic, while individually within each nanoparticle, antiferromagnetic, ferrimagnetic or ferromagnetic ordering of magnetic moments can be realized. Based on the temperature dependence of magnetization, it can be assumed that all of the above scenarios are realized, since it is impossible to unambiguously describe this dependence within the framework of one model. The blocking temperature for the ABS+6%[PEG/CoB-NP] sample was 13 K. In turn, any of these three scenarios is consistent with the superparamagnetic behavior of the nanoparticle ensemble."

should be reduced to

"Based on the field and temperature dependence of the sample magnetization (Figure 6), the magnetic state of the resulting composite ensemble can be defined as superparamagnetic. The blocking temperature for the ABS+6%[PEG/CoB-NP] sample was 13 K."

The discussion about the possible magnetic state of the individual nanoparticles needs further investigation has nothing to add to the manuscript at the moment.

Author Response

I want to thank the authors for addressing my comments regarding their manuscript. The present work would be suitable for publication after the implementation of two minor alterations.

On behalf of all the authors of the work, I thank the reviewer for the efforts aimed at improving our manuscript.

  1. A thorough proofreading in the captions before publication. There are still some typos (Figure 4) and words missing (Figure 6).

Response: We once again thank the reviewer for the omissions indicated. Captions for all figures have been added and brought into uniformity.

  1. The following passage

"Based on the field dependence of the sample magnetization (Figures 6C and 5E), the magnetic state of the resulting composite ensemble can be defined as superparamagnetic, while individually within each nanoparticle, antiferromagnetic, ferrimagnetic or ferromagnetic ordering of magnetic moments can be realized. Based on the temperature dependence of magnetization, it can be assumed that all of the above scenarios are realized, since it is impossible to unambiguously describe this dependence within the framework of one model. The blocking temperature for the ABS+6%[PEG/CoB-NP] sample was 13 K. In turn, any of these three scenarios is consistent with the superparamagnetic behavior of the nanoparticle ensemble."

Should be reduced to

"Based on the field and temperature dependence of the sample magnetization (Figure 6), the magnetic state of the resulting composite ensemble can be defined as superparamagnetic. The blocking temperature for the ABS+6%[PEG/CoB-NP] sample was 13 K."

The discussion about the possible magnetic state of the individual nanoparticles needs further investigation has nothing to add to the manuscript at the moment.

Response: We agree with this remark. The discussion of magnetic properties, in the context of this work, was somewhat speculative. The appropriate changes suggested by the reviewer have been made to the text.

On behalf of all the authors, I would like to thank the reviewer once again for his work and time spent.
Taking this opportunity, I would like to congratulate you on the upcoming New Year and wish you success in the coming year.

Reviewer 2 Report

Comments and Suggestions for Authors

The manuscript has been revised properly, I suggested it can be accepted in the present form.

Author Response

On behalf of all the authors, I thank the reviewer for his work and time spent.
Taking this opportunity, I would like to congratulate you on the upcoming New Year and wish you success in the coming year.